# Deep recurrent reinforced learning model to compare the efficacy of targeted local versus national measures on the spread of COVID-19 in the UK

Tim Dong ![ORCID] , Umberto Benedetto, Shubhra Sinha, Daniel Fudulu, Arnaldo Dimagli, Jeremy Chan, Massimo Caputo, Gianni Angelini

TD and UB contributed equally.

Bristol Heart Institute, Bristol Medical School, University of Bristol, Bristol, UK

**Correspondence to**
Dr Umberto Benedetto;
umberto.benedetto@bristol.ac.uk

## ABSTRACT

**Objectives** To prevent the emergence of new waves of COVID-19 caseload and associated mortalities, it is imperative to understand better the efficacy of various control measures on the national and local development of this pandemic in space–time, characterise hotspot regions of high risk, quantify the impact of under-reported measures such as international travel and project the likely effect of control measures in the coming weeks.

**Methods** We applied a deep recurrent reinforced learning based model to evaluate and predict the spatiotemporal effect of a combination of control measures on COVID-19 cases and mortality at the local authority (LA) and national scale in England, using data from week 5 to 46 of 2020, including an expert curated control measure matrix, official statistics/government data and a secure web dashboard to vary magnitude of control measures.

**Results** Model predictions of the number of cases and mortality of COVID-19 in the upcoming 5 weeks closely matched the actual values (cases: root mean squared error (RMSE): 700.88, mean absolute error (MAE): 453.05, mean absolute percentage error (MAPE): 0.46, correlation coefficient 0.42; mortality: RMSE 14.91, MAE 10.05, MAPE 0.39, correlation coefficient 0.68). Local lockdown with social distancing (LD_SD) (overall rank 3) was found to be ineffective in preventing outbreak rebound following lockdown easing compared with national lockdown (overall rank 2), based on prediction using simulated control measures. The ranking of the effectiveness of adjunctive measures for LD_SD were found to be consistent across hotspot and non-hotspot regions. Adjunctive measures found to be most effective were international travel and quarantine restrictions.

**Conclusions** This study highlights the importance of using adjunctive measures in addition to LD_SD following lockdown easing and suggests the potential importance of controlling international travel and applying travel quarantines. Further work is required to assess the effect of variant strains and vaccination measures.

## INTRODUCTION

COVID-19 is a highly infectious disease that resulted in a global pandemic in just under a month.[1] This pandemic has caused global disruptions to individuals, businesses and

### Strengths and limitations of this study

⇒ The proposed deep recurrent reinforced learning (DRRL) based model takes into account of both relationships of variables across local authorities and across time, using ideas from reinforcement learning to improve predictions.

⇒ While predicting the geographical trend in COVID-19 cases based on the simulation of different measures in the UK at both the national and local levels in the UK has proved challenging, this study has provided a methodology by which useful predictions and simulations can be obtained.

⇒ The Office for National Statistics only released data on UK international travel up to March 2019 at the time of this study, and therefore, this study used the amount of UK tourists in Spain as a reference variable for understanding the effect of international travel on COVID-19 spread.

governments worldwide. The number of cases has continued to rise exponentially, from 80,239 in February 2020 to 69 million as of December 2020.[1] COVID-19 is unlike other historic pandemics in terms of its rapid worldwide spread, a substantial increase in infected and symptomatic people and a rapid development of newly evolving strains. Recent cases of a new variant of COVID-19 have also been found.[2] These problems are being faced worldwide despite global efforts to control this virus.

The spread of COVID-19 can be modelled as a four-stage process: (1) appearance of disease; (2) local transmission; (3) community transmission; and (4) epidemic outbreak.[3] An area can be defined as a liberal zone, a surveillance zone or an infected zone, depending on regional infection patterns, and different levels of restriction measures can be applied.[3] Studies have also focused on the effect of quarantine on COVID-19 spread and have found that it is more effective than

control and combinations with other measures, for example, school closures, travel restrictions and social distancing had a synergistic effect.[4]

Epidemic models developed so far have aimed at understanding the effect of various quarantine factors and mostly applied Newtonian calculus approaches.[5] One study modelled the strictness of lockdown interventions using a contact factor F (ranging from 3 to 8), with three being the strongest and eight being the weakest.[6] In another study, researchers used a COVID-19 decision-making system based on differential formulas and stochastic methods to model transitions between population phase states such as susceptible, exposed, infected, hospitalised, recovered and died. The study was extended to incorporate demographics and social status variables using data from official statistics and the literature.[7] In time-series data analysis and forecasting, deep learning (DL) shows promise. DL models can automatically learn temporal connections and patterns in the data, such as trends and seasonality.[8] Time series and geographical data analysis have been applied to study and inform on optimal energy sector management policies to mitigate the effect of COVID-19.[9] Another study also visualised the geographical distribution of COVID-19 cases.[10] For forecasting worldwide COVID-19 incidence as well as for country-specific and city-specific predictions, one study employed statistics measures to sort the most effective model for medium-term prediction using ARIMA, LSTM, Stacked LSTM (SLSTM) and Prophet models.[11] NAR and FITNET neural networks were combined as an ensemble using a fuzzy weighted approach to predict 10 days ahead of 12 Mexican states.[12] Fuzzy rules have been applied, along with fractal dimension as transformation criteria, to account for linear and non-linear dimensionality in order to forecast the COVID-19 trend.[13] Following this approach, expert knowledge was used to define rules and class memberships with a different set of countries.[14] To model the effect of control measures, a control loop system was used with a novel set of fuzzy logic, with the error between the observed and desired number of infections and the linear fractal dimension of the country as input.[15]

### Research gap

From the literature review, we can determine that there are a range of time-series prediction models, each of which outperforms in distinct situations and has its own set of limitations. Although LSTM variants have been used, there have been limited reports of the gated recurrent units (GRUs) DL model. In addition, no DL model results have been mapped to a two-dimensional (2D) choropleth map in order to visualise the effect of control measures. Besides, no application of reinforcement theory has been used for DL analysis. Furthermore, the DL models have not been linked directly to the government website. To add to this, there is a lack of DL models that apply a combination of expert designed matrices and official statistics/government data to incorporate social

demographic risk factors for modelling the effects of implementing various restriction measures.

In order to address the limitations of the existing system, the proposed work focuses on the analysis mapping of results from a reinforcement-based DL GRU model (trained with data including longitude and latitude coordinates) onto 2D choropleth maps in order to understand the effectiveness of various control measures. The proposed model is also linked to the Government UK website and an expert-curated matrix to incorporate effects of control measures and social demographic risk factors.[16] A web dashboard for the DL model was built. To the best of our knowledge, this is the first study to apply these techniques to include the examination of hotspot (high incidence) areas in the UK.

Here, the proposed work examines the 2D geographic trend based on simulations of various control measures at both the national and local authority (LA) levels in the UK in order to have a detailed understanding of the factors affecting the spread of COVID-19 at these levels as well as the potential impact of future policy measures. This knowledge would allow the UK government, LAs and individual citizens to make informed decisions about regional policies and personal exposure risks.

## METHODS

### Patient and public involvement

This research was done without patient and public involvement.

### Model development

The proposed model enables predictions of the incidence and mortality related to COVID-19 in the upcoming 5 weeks and simulates the effect of control measures targeting the COVID-19 spread, that is, the number of facilities available for accommodation and food, pubs, retail shops, education, transport and storage, art, entertainment and recreational services, within each LA region. The model also accounts for international migration inflow, internal migration inflow and outflow within the UK, thus simulating control measures that affect travel.

The proposed model is a deep recurrent reinforced learning (DRRL) based model (online supplemental material Part I) named National Coronavirus Global Forecast System (NCGFS) that combines the synergistic properties of GRU[17] and reinforcement deep learning.[18] Like other DL models, GRU has the ability to model non-linear and temporal relationships between and within high dimensions of variables. However, GRU is also expected to be well suited in small dataset scenarios and is computationally more efficient.[19] The reinforcement learning element of NCGFS enables it to adapt to newly inputted data and make more accurate forecasts.

All available LA data were split 80:20 into training and validation data subsets. Data were preprocessed using scaling—subtracting their corresponding mean and dividing by the SD values. Following the completion of

predictions, the prediction outputs are then scaled back to their original scale.

The NCGFS neural network model used an input layer, numerous hidden layers and an output layer. A complex series of non-linear matrix computations are applied to the input data to relate the target output (ie, cases and mortality) and to the other data columns (eg, amount of international migration inflow or internal migration inflow and outflow within UK, number of retail shops, etc). The model is first trained using the existing data subsets provided through a data generator. Each column of the data is assigned a specific weight at each of the nodes in the hidden layers, and these weights are progressively updated to minimise the mean absolute error (MAE) between the predicted and actual values, using the RMSProp optimisation algorithm.[20] Selection for RMSProp is further detailed in online online supplemental material Part I, recurrent optimisation algorithms. During the prediction process, an input data matrix of the same dimension as the training data is then passed into the input layer. The neural network's hidden layers then use the weights learnt during the training process to predict the most likely incidence and mortality based predictor variables from each corresponding week.

The final model consists of two components, model-M (master model) and model-R (reinforced model) that serve different purposes. Model-M accounts for the relationships of variables across different LAs, while model-R provides improved forecasting performance for each individual LA that are selected for analysis. For detailed specifications of model parameters, please see online supplemental material, part I neural network architecture and configuration. This model is particularly apt at generalisation and is capable of forecasting a wide range of LA simultaneously. The model uses model-R to increase forecast performance for the individual LA that are selected for analysis. Model-M is updated with several additional epochs of training data from the selected LA to reinforce and optimise the predictions. Software code is available through https://githubcom/s0810110/Cvd_NCGFS_TrendAnalysis.

### Data linkage

The data used to train the deep learning model is based on various datasets that have the potential to influence the trend in the number of COVID-19 cases and mortality at the national and local level (refer to online supplemental materials, part II. A for more details), including domains of deprivation,[21] number of bars and pubs,[22] business size,[23] population estimate (male, female, by age and overall),[24] etc.[25–27] We use the R language and the R software development kit (SDK) for COVID-19, that is, a set of software commands to retrieve data remotely, as published by Public Health England, to automatically extract the latest daily cases and mortality figures for all LA within the UK. Using this approach, we are able to automate and dynamically predict the cases and mortality as new data are generated by GOV.UK. We use R to convert these data from daily figures into weekly counts and link these data to the data described in online supplemental materials, Part II. A.

Specifically, knowledge from experts in risk modelling is used to curate a matrix containing three indices that together are named the COVID-19 General Policy indices: LockdownScore, QuarantineMeasures[28] and SchoolOpening.[27 29] The main dataset is connected to these index scores based on the weeks each of the associated policies was implemented and the relative effects at each time period.

Furthermore, the number of tourists arriving in Spain from January 2020 to July 2020 were obtained and adjusted by the proportion of UK tourists in Spain from the year 2019. As data on international travel is not readily available for the period affected by COVID-19, the rationale is to use the amount of UK travel to Spain as an indicator for the impact of international travel on the spread of COVID-19,[30 31] since Spain is a frequent UK tourist destination. Our GRU model is trained on the above data and includes the longitude and latitude of each LA as part of the model.[32]

A secured web dashboard was developed that enables users to explore the adjustment effects of risk factors and control measures on the spread of COVID-19 and can be made available on request (http://137.222.198.54:8081/).

While the LA boundaries data are not included in the training process, the main dataset is also linked to these data following forecast generation, so the deep learning model will also provide the prediction of the incidence or mortality in the next 5 weeks in a geographical map view. Furthermore, the model has the capability to toggle the map view by LA or Public Health England regions. These views will be useful for the government to see the future effects of different control measures changes and for the individual citizens to understand their risk of movement within and between local regions in the upcoming future.

For analysis in the map view, the geographical regions from the top to bottom of England is divided into four equidistant slices, which we shall name slice n2, n1, s1 and s2, respectively. These categories will be applied to all other geographical plots hereafter to facilitate discussion. The areas with a higher number of cases are shown in darker colours with 6 grades of severity (I–VI) covering the ranges 0–250 (I); 250–500 (II); 500–750 (III); 750–1000 (IV); 1000–1250 (V); and 1250–1500 (VI). Any number outside of this range is shown in grey and is classed as grade VII.

### Model validation

The model is internally validated for the whole of England, whereby the model is trained using all data except for weeks 41–46. The data from this interval are evaluated using root mean squared error (RMSE), mean absolute error (MAE), mean absolute percentage error (MAPE) and correlation coefficient.[11 12] Average ranking of performance metrics were performed as per eqs. 24 and 25.[11]

At the time of this work, only data up to week 46 are available. The risk and control parameters are adjusted within the web dashboard from week 40 onwards to enable predictions to simulate a full/national lockdown (FLD) from week 45 onwards. This is because it is known that a FLD had been applied in the UK from week 45 (00.01 on Thursday, 5 November 2020), and prior to that, local lockdown with social distancing (LD_SD) had been implemented.[33]

During the revision of this work, data for week 51 had become available and was downloaded from GOV.UK to enable external validation of simulated results. This was performed for top 21 hotspots using LD_SD and FLD separately. The same statistic metrics were used as that for internal validation. The following section explains the model simulation process in more details.

### Model simulation

Simulations are performed using the final model that is trained using the approach described in the model development section. All data, that is, from week 5 to 46 are included for training this model. The model is used to simulate the effects of numerous different COVID-19 prevention measures on the number of cases at week 51, that is, 5 weeks ahead of the latest available data. The risk and control parameters that model the corresponding measures are set from week 40 onwards to enable predictions to simulate the implementation of various measures from week 45 onwards, rather than FLD, which was what the government actually implemented. The measures simulated are: (A) No lockdown versus LD_SD; (B) LD_SD versus FLD; (C) LD_SD versus LD_SD with international travel −50%; (D) LD_SD versus LD_SD with closing school −50%; (E) LD_SD with travel quarantine 5.5 (see online supplemental material, part II. A., 11) versus LD_SD with full travel quarantine 10; (F) LD_SD with 100% pubs open versus LD_SD with −50% pubs; (G) LD_SD with 100% food and accommodation services open versys LD_SD with −50% food and accommodation services open; (H) LD_SD with −50% retail services open versus LD_SD with 100% retail services open. For details on the implementation of these measures, please refer to online supplemental material, part II. A.

These measures are simulated first for individual LA by selecting a baseline LA with a relatively low case count and comparing the effect of the measures when applied to a LA with a very high number of cases, that is, a hotspot area. The measures are then ranked by order of effectiveness. This is so that the relative effectiveness of each measure can be understood at the local level. Second, the measures are simulated for all the LA in England to visualise the relative effectiveness of each measure at a national level. For the 21 LAs with the highest cases when using a LD_SD measure, the predicted cases counts at week 51 are extracted and plotted to analyse the efficacy of each measure across these nationally 'hard' to tackle areas. This comparison also enabled the ranking of the relative effectiveness of each measure at these hotspots.

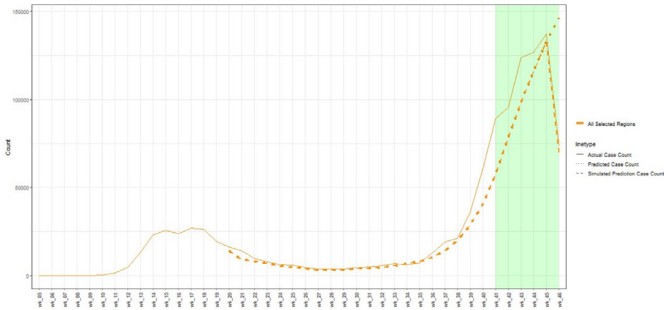

**Figure 1** Validation of cases for week 46 with weeks 41–46 excluded from data.

### RESULTS

Model validation of predictions against actual results for week 46 showed a good match between the simulation and an actual number of cases across all the LA concerned (figure 1). The model distinguished the LA with high cases from the areas with a low number of cases (figure 2A,B). Furthermore, the model performs especially well for low-grade LA (table 1). The tendency towards better performance in low degree LA, maybe because data from weeks 41 to 46 containing sharp changes in the trend have not been included. Good performance was achieved in terms of RMSE, MAE, MAPE, correlation coefficient and ranking when benchmarked against Devaraj *et al*[11] and Melin *et al*[12] (online supplemental tables S2 and S3, part III). FLD simulation performed better than LS_SD in external validation using the top 21 hotspots. Results from

**A** Actual cases Wk 46

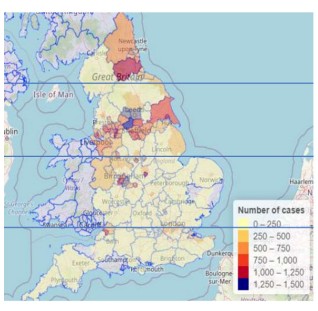

**B** Validation of cases Wk 46

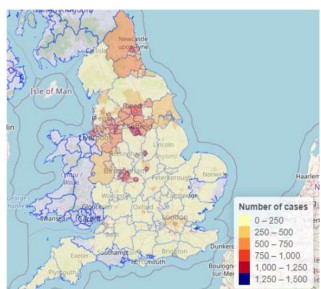

**C** LD_SD Wk 51

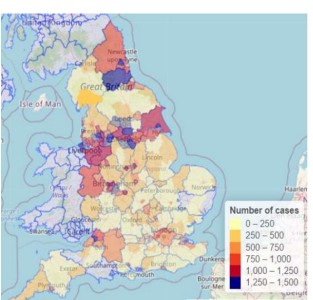

**D** FLD Wk 51

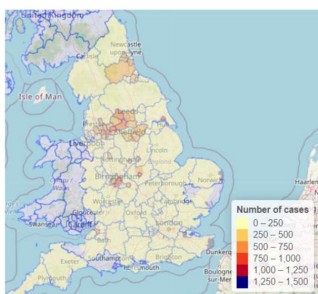

**Figure 2** Geographical level of cases for actual and predicted results based on different measures. (A) Exemplifies the use of geographical slices n2, n1, s1 and s2. Additional results are available in online supplemental materials, part II. B. FLD, full/national lockdown; LD_SD, local lockdown with social distancing.

**Table 1** Validation model: number of actual and predicted cases and mortalities

| Local authority | Number of actual cases for week 46 | Cases forecast for week 46 | Number of actual mortalities for week 46 | Mortality forecast for week 46 |
|---|---|---|---|---|
| Intervention | | *Full lockdown* | | |
| Wolverhampton | 438 | 482 | 4 | 5 |
| Gedling | 179 | 196 | 6 | 2 |
| Welwyn Hatfield | 119 | 130 | 0 | 2 |
| Wiltshire | 201 | 219 | 1 | 4 |
| Portsmouth | 220 | 239 | 0 | 3 |
| Bromley | 217 | 232 | 2 | 3 |
| Stockton-on-Tees | 467 | 498 | 7 | 7 |
| Stockport | 517 | 550 | 13 | 8 |
| South Kesteven | 153 | 162 | 6 | 1 |
| Hammersmith and Fulham | 166 | 175 | 1 | 2 |
| Kingston upon Thames | 150 | 158 | 4 | 2 |
| Ribble Valley | 93 | 98 | 4 | 2 |
| East Cambridgeshire | 34 | 36 | 1 | 1 |
| Redcar and Cleveland | 380 | 396 | 7 | 4 |
| Sedgemoor | 55 | 57 | 3 | 1 |
| Cheshire East | 496 | 514 | 6 | 7 |
| Wealden | 76 | 79 | 1 | 2 |
| Charnwood | 371 | 382 | 3 | 3 |
| South Somerset | 72 | 74 | 1 | 1 |
| Southend-on-Sea | 137 | 140 | 0 | 3 |
| Chelmsford | 110 | 112 | 2 | 2 |
| Rushcliffe | 124 | 126 | 4 | 2 |
| Merton | 146 | 148 | 0 | 2 |
| Shropshire | 426 | 428 | 6 | 6 |
| Harrogate | 253 | 253 | 1 | 2 |
| Central Bedfordshire | 226 | 225 | 5 | 4 |
| Sutton | 155 | 154 | 5 | 3 |
| Oldham | 735 | 732 | 15 | 8 |
| Hillingdon | 325 | 323 | 3 | 3 |
| Basildon | 168 | 167 | 4 | 3 |
| Plymouth | 196 | 192 | 2 | 3 |
| Test Valley | 59 | 58 | 1 | 1 |
| Walsall | 605 | 590 | 15 | 6 |
| Southampton | 187 | 182 | 0 | 2 |
| Selby | 129 | 124 | 2 | 1 |
| South Holland | 105 | 100 | 2 | 1 |
| Chiltern | 57 | 54 | 0 | 1 |
| Derbyshire Dales | 82 | 78 | 1 | 2 |
| Chichester | 58 | 54 | 0 | 1 |
| Barnet | 378 | 354 | 5 | 4 |
| Tameside | 447 | 417 | 18 | 12 |
| Salford | 577 | 537 | 17 | 7 |
| Havant | 77 | 71 | 1 | 1 |
| Waverley | 97 | 89 | 0 | 1 |
| Nuneaton and Bedworth | 247 | 226 | 4 | 3 |
| New Forest | 92 | 84 | 6 | 1 |
| Ryedale | 67 | 61 | 1 | 1 |

**Table 1** Continued

| Local authority | Number of actual cases for week 46 | Cases forecast for week 46 | Number of actual mortalities for week 46 | Mortality forecast for week 46 |
|---|---|---|---|---|
| Peterborough | 224 | 204 | 2 | 4 |
| North Hertfordshire | 89 | 81 | 1 | 1 |
| Epping Forest | 130 | 118 | 2 | 1 |

The results show that there is a close match between the actual and predicted number of cases, especially for LA at grade III or below.
Only 50 LAs are displayed. For validation data on all LA, please contact the authors.

simulation of cases and mortality up to week 51, using data from weeks 41 to 46, are further discussed further.

The effects of different measures were first observed at a local level. Southampton was selected as a baseline for observing the effects of measure changes. As Southampton is a grade I LA with a low case number of 187 in week 46, the effects of measure changes were readily perceived with effectiveness ranked from most effective to least effective (online supplemental figure S6, part IV): (b) full lockdown; (c) LD_SD and international travel −50%; (e) LD_SD and 100% quarantine; (d) LD_SD and closing school −50%; and (f) LD_SD and closing pubs −50%. There were negligible differences observed between LD_SD and (g) LD_SD & −50% food and accommodation and (h) LD_SD and −50% retail.

As Leeds was in the highest grade (VII) for both week 46 (actual) and week 51 (predicted), it was selected for observing the effects of different measures on 'hard' to tackle areas. As the number of cases for Leeds was approximately five times higher than Southampton, the effect of measures relative to the number of cases in any week were much smaller in the former than the latter. For Leeds, no difference was observed for predicted cases at week 51 between no lockdown and LD_SD. Full lockdown (online supplemental figure S7b, part IV) was the most effective, followed by LD_SD with a reduction in international travel by 50%, although the effects were much less in proportion to the number of cases than Southampton. There was a negligible impact on the number of cases at week 51 for the remaining measures (online supplemental figure S7e-h).

Figure 2C shows the predicted cases in week 51 using LD_SD. At a national level, it can be seen that there would be a rapid rise in the number of cases, especially in the horizontal 'belt' along the n1 region. In addition, there is at least one LA in each of the other slices n2, s1 and s2 that are expected to rise to grade VI or above. The majority of LA locations elsewhere, which were mostly at grade I in week 46, are expected to rise to grade II or III. The top 21 hotspots at week 51 using LD_SD were selected for subsequent analysis (table 2).

LD_SD was shown (figure 3) to be effective in suppressing the increase in cases for Birmingham (−17%), Bradford (+0.98%), Kirklees (−6.6%) and Leicester (−1.3%). LD_SD was shown to be ineffective for suppressing the increase in cases for the remaining 17 LA, with the highest predicted rises for Wirral (325%),

Stockport (163%), Tameside (188%), Rotherham (158%) and Derby (130%).

LD_SD with −50% international travel was the most effective measure after full lockdown (blue vs brown, figure 4). One hundred per cent quarantine (pink) was the next most effective supplementary measure, with similar effectiveness to international travel −50% except for three LA. Notably, LD_SD with 100% quarantine resulted in higher cases than LD_SD with international travel −50% for Bradford (+9.1%) and Leicester (+7.6%). As an exception, Manchester had −41% fewer cases when using the quarantine measure compared with international travel restrictions.

The supplementary effect of school closing −50% was less than international travel restrictions for all 21 LA, with the number of cases being (+9.2%) higher on average using the former measure. Closing pubs −50% had a similar, although slightly lower level of effectiveness compared with school closing, with a higher number of cases (+2.2%) on average using the former measure compared with the latter. Again, reducing the number of food and accommodation services, −50% had a similar but a slightly lower level of effectiveness compared with pubs closing, with the number of cases (+2.0%) being higher on average using the former measure. In addition, a reduction in the number of retail services −50% resulted in a similar effect to food and accommodation services −50%, with on average a minimal increase in the number of cases (+0.29%) using the former measure. It can be seen that, on average, the ranking of measure effectiveness for the national hotspots are the same as the local baseline, that is, Southampton.

## DISCUSSION

Previous studies have evaluated the prediction performances of DL and non-DL based models without 2D choropleth analysis and found that SLSTM outperformed other models because of better hyperparameter tuning and reduction in bias. In addition, it was found that ARIMA outperforms LSTM model.[11] Using the same average ranking metrics, we found that NCGFS (overall rank 1–3) outperformed SLSTM (overall rank 4), ARIMA (overall rank 5) and LSTM (overall rank 6). However, there are several limitations of this comparison: (1) predictions are for different countries; (2) lower mortality rates in England compared with India may bias better ranking towards NCGFS; and (3) comparison does not account for recovered cases. NCGFS also demonstrated

**Table 2** Final model: number of actual and predicted cases and mortalities

| Local authority (LA) | Number of actual cases for week 46 | Number of actual mortalities for week 46 | Cases forecast for week 51 | Mortality forecast for week 51 | Cases forecast for week 51 | Mortality forecast for week 51 |
|---|---|---|---|---|---|---|
| Intervention | *Full lockdown* | | *Local lockdown with social distancing* | | *Full lockdown* | |
| Leeds | 1801 | 17 | 1881 | 21 | 499 | 17 |
| Sheffield | 948 | 36 | 1784 | 32 | 275 | 14 |
| Birmingham | 1957 | 35 | 1627 | 25 | 537 | 17 |
| Wigan | 759 | 34 | 1554 | 24 | 346 | 10 |
| Manchester | 1067 | 13 | 1550 | 20 | 427 | 13 |
| Bradford | 1534 | 24 | 1549 | 20 | 722 | 17 |
| Stockport | 517 | 13 | 1529 | 17 | 218 | 7 |
| Liverpool | 750 | 29 | 1509 | 22 | 234 | 8 |
| Rotherham | 561 | 20 | 1448 | 22 | 257 | 7 |
| Kingston upon Hull | 1011 | 21 | 1368 | 18 | 584 | 12 |
| Oldham | 735 | 15 | 1336 | 17 | 509 | 11 |
| Wirral | 311 | 13 | 1324 | 19 | 65 | 4 |
| Bolton | 635 | 18 | 1299 | 17 | 447 | 11 |
| Bristol | 763 | 8 | 1296 | 14 | 444 | 13 |
| Tameside | 447 | 18 | 1288 | 20 | 255 | 7 |
| County Durham | 1161 | 23 | 1252 | 26 | 377 | 7 |
| Derby | 537 | 11 | 1234 | 15 | 335 | 8 |
| Walsall | 605 | 15 | 1233 | 21 | 288 | 7 |
| Kirklees | 1292 | 25 | 1207 | 29 | 557 | 14 |
| Leicester | 1006 | 7 | 993 | 15 | 581 | 11 |
| Sandwell | 762 | 21 | 969 | 25 | 398 | 10 |

Results are shown for the top 21 LA with the highest predicted cases observed at week 51 using LD_SD.
LD_SD, local lockdown with social distancing.

good performance in terms of case prediction when benchmarked against Modular Neural Network with Fuzzy (MNNF) in terms of RMSE (700.88 vs 1554.03).[12] While a few studies have analysed 2D geospatial predictions, for example, by converting to heatmaps,[34] and considered hotpot regions, these have mainly been using either modified regression or differential equation techniques rather than DL-based techniques.[10 35 36] Although interactive dashboards have been developed for tracking COVID-19,[37] these typically do not enable prediction through simulation of control measures. Furthermore,

although reinforcement learning has been applied, it has not typically been combined with 2D map analysis or lack external validation.[38 39] In this article, we demonstrate the use of a reinforcement-based DL GRU model with 2D choropleth maps to analyse spatial representation of results in order to rank the efficacy of various control measures. The proposed model is embedded in an interactive dashboard linked to the Government UK website and an expert-curated matrix to incorporate effects of control measures and social demographic risk factors. This combination of techniques has not yet been

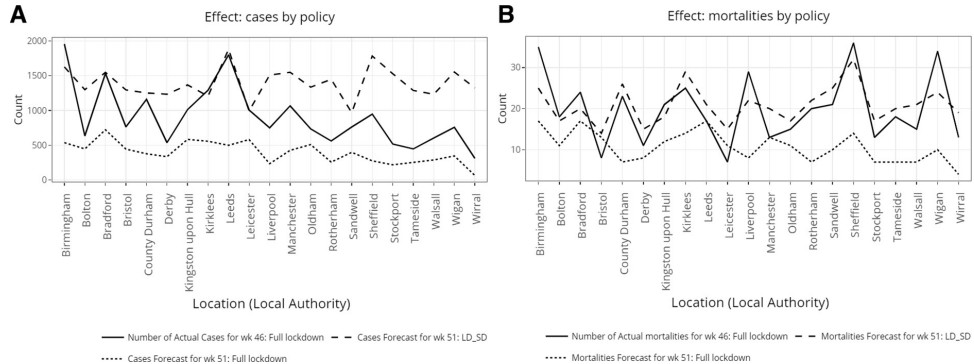

**Figure 3** For the top 21 LA with the highest predicted cases observed at week 51 using LD_SD, plots were generated to compare the effects of full lockdown against LD_SD in terms of cases (A) and mortalities (B). LA, local authority; LD_SD, local lockdown with social distancing.

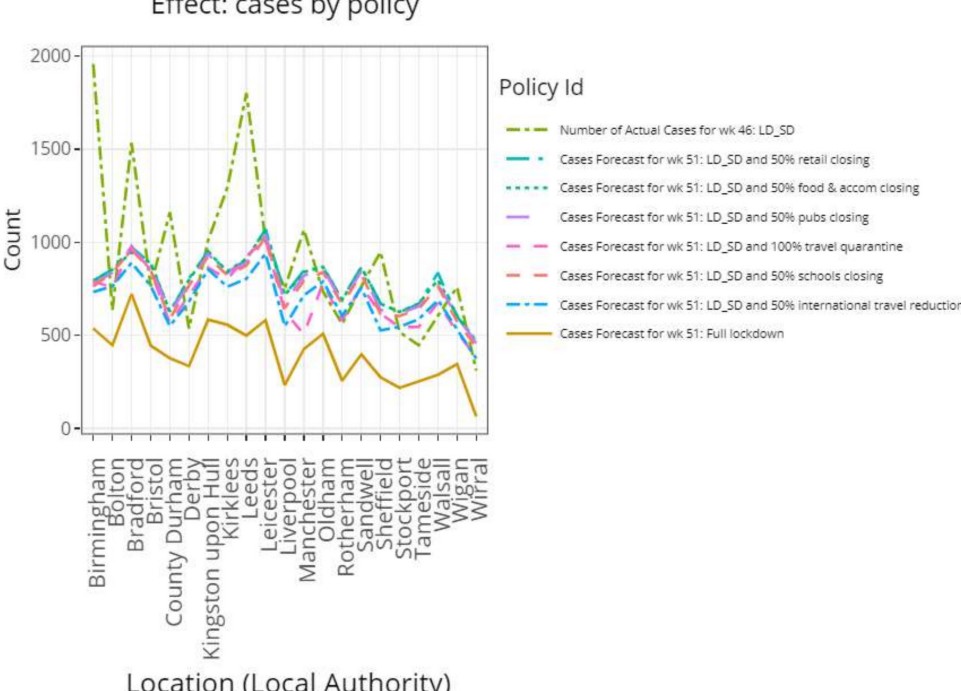

**Figure 4** For the top 21 LA with the highest predicted cases observed at week 51 using LD_SD, a plot is generated to compare the effect on the number of cases using a combination of LD_SD with other 'supplementary' measures. LA, local authority; LD_SD, local lockdown with social distancing

widely used, and it provides a number of advantages over each individual formulation alone. The NCGFS model can be used to make inferences into the effectiveness of different measures at both the national and local levels. The model suggests that there is variation in the effect of each measure across different regions. Notably, our results indicate that the protective effects of lockdown measures benefit some local authorities more than others (online supplemental material fig S6 and S7, part IV) and that local lockdown with social distancing is ineffective compared with national lockdown in suppressing the increase in cases for most of the LA areas. That is, if the government had kept the same local lockdown with social distancing policies, which they had implemented from week 40 onwards rather than switching to a national lockdown policy at week 45, then we would have seen a rapid rise in cases nin the n1 belt region and in areas such as County Durham (n2), Bristol (s2) and Birmingham (s1), as well as in many other areas across England.

Local lockdown with social distancing and without additional measures may be inefficient in stopping rapid rise of hotpot regions due to the geographical properties of hotspot regions. Hotspots along the middle of the n1 geographical slice constitute a tight cluster of large metropolitan cities. The high number of cases may be partly attributed to the high number of services such as pubs and schools available and the amount of travel in these areas. We expect that this effect could possibly be enhanced by the fact that the n1 slice contains a large number of LA areas with many boundary links to other hotspots, which agrees with another study that highlighted influences of

neighbouring small areas and found continuous bands of hotspot regions.[36]

Since the government is only able to impose a national lockdown for a limited period, follow-up measures should incorporate LD_SD with additional measures as LD_SD alone is likely not to be sufficient. The actual FLD had lasted only up to 2 December 2020 (week 49), after which LD_SD was implemented by the government. The UK Government had implemented LD_SD for 2 weeks in between week 49 and week 51. As a substantially larger proportion of weeks between week 45 and week 51 were implemented using FLD rather than LD_SD, we found the FLD simulation (overall rank 2) better predicted actual results than LD_SD (overall rank 3) as expected. Trend and 2D map analysis showed that introduction of additional measures on top of local lockdown with social distancing can help to suppress the increase of or even decrease the number of cases in national hotspots as well as local areas where cases are not very high. Our model shows that the ranking of the average effectiveness of each supplementary measure is consistent across the national hotspots and local baseline, and this ranking can be used to prioritise those interventions according to an order of effectiveness. Nonetheless, it was also observed that specific measures are more effective for some LA compared with others. In these cases, it is necessary to adjust the priorities of the measures implemented accordingly.

The model has highlighted the importance of reducing the amount of international travel, the number of open schools and pubs as well as the implementation of travel

quarantine procedures in controlling the spread of COVID-19 over other measures, such as reducing the number of food and accommodation and retail services, which seemed less relevant on the virus spread (figure 4 and online supplemental figure S6). Our finding of the usefulness of restricting international travel and applying travel quarantine is in contrast with another study, which found that quarantine of travel from endemic countries was not effective.[4] One explanation for international travel being more impactful than travel quarantine is that while travel quarantine provides the government with some control over which countries to enforce a 14-day self-isolation on travellers' return, it provides little control over the activities of travelling individuals once they arrive at their destinations as well as the level of preventative health measures at those destinations. Furthermore, the individuals who travel are more likely to encounter places of gathering while abroad. In addition, even with COVID-19 testing in place, the journey from the airport back to the home of the traveller allows an increased opportunity of spreading the virus, particularly if public transport such as taxis or buses are used.[4 8] Therefore, these measures are not as direct as limiting the amount of international travel.

While closing schools were not as effective as international travel and quarantine restrictions, we found this measure to be more effective than closing pubs. One potential explanation for this is that schools are more crowded places and are subject to a more frequent number of close contact scenarios in comparison with the pub. The view that schools contribute to the spread of COVID-19 has been supported by the literature.[40 41] While the virus may pose a low risk of mortality to the children themselves, these frequently asymptomatic carriers can also lead to the spread of the virus to their households, teachers and communities.

The reason why minimal effects were found for food and accommodation and retail restrictions may be because in these sectors, people generally associate with others that they are closely associated with. For example, families are more likely to sit with each other in restaurants or walk together when shopping rather than with people they are less familiar with. This is not the case in pubs as anyone from the communal area can be present.

It is unexpected that in the s2 slice that Bristol has higher predicted cases than the LA in the London area as one would have thought the latter comprising a total population of 9 million (2019) and a high traffic volume owing to its large underground network system would result in much higher case numbers. We expect that this may be because the London region LA generally has less health and disability deprivation (deciles: Wandworth: 7; Barnet: 8.9; Brent: 7.3; and Waltham Forest: 6.1) compared with Bristol (decile: 4.4). This is supported by the findings that suggest that existing comorbidities are associated with an increased likelihood of COVID-19 hospital admission.[42]

In light of evidence given by the comparison between the LA within the London region and Bristol, we expect

the effect of LA boundary connections to be adjusted by the degree of health and disability deprivation. Indeed, we found that regions with a high number of cases along the horizontal 'belt' in the n1 region had a high degree of health and disability deprivation (decile: Manchester: 1.9; Leeds: 4.1; Bradford: 3.3; Liverpool: 1.8; Sheffield: 3.9; and Wigan: 3.4). Another study reports a similar result.[36] This also applies to County Durham (n2), which has a high degree of health and disability deprivation (decile: 2.9) and was seen to have a significant increase in cases at week 51 using a LD_SD measure.

One limitation of our study is that for simulated week 51, there was an LA in Kent that was coloured grey on the map but were, in fact, reporting negative values. A future improvement would be to transform or limit the prediction outputs to retain meaningful information while preventing negative values. Nonetheless, the result is interesting and giving the LA a grade of VII may still be valid, since the highly infectious Kent variant emerged in week 39 (20 September).[43] The present study has not specifically dealt with the modelling of variants. In addition, vaccination effects could not be accounted for as these were only beginning to be rolled out (2 December 2, week 49) at the point of this study.[44] While a variety of data sources have been used, this study has not analysed the effect of in-hospital admission or clinical comorbidity. These are issues that deserve to be further explored. Future work could be done to compare the current model against those found in similar studies, for example, ARIMA, LSTM, SLSTM and Prophet,[11] Bayesian hierarchical space–time SEIR model,[36] NAR and FITNET neural networks, using the same dataset.[12] Furthermore, a Hybrid Q-learning based algorithm could be used whereby these models represent potential actions to update the Q cumulative reward matrix.[39] While the current model takes a risk factor matrix as one of its inputs and this was built with expert input, a fuzzy logic approach with functionally modelled inputs and outputs was not used.[15] Incorporating such methods could enhance the interpretability of the risk factor matrix.

## CONCLUSION

This study highlights the importance of simulating the effects of various control measures using map and non-map-based analyses to prioritise COVID-19 preventative measures. This was demonstrated for both local hotspot zones and on a nationwide scale. Furthermore, at the LA level, we demonstrated the utility of geographical slicing for comparative analysis of interventional effects across time periods and thereby can also allow governments to assess the optimal measures to apply. It is advisable to assess the effectiveness of lockdown with social distancing alone against that when combined with other adjunctive measures and implement periodic monitoring at both trend and map dimensions to reduce the risk of outbreak rebound following lockdown easing. Lastly, this study highlights the importance of controlling international

travel, and this should be further explored with the comparative analysis of effectiveness against newly developed vaccine measures.

**Contributors** TD and UB conceived and designed this study. TD and UB acquired the data. TD, UB, SS, DF and AD analysed and interpreted the data. MC, DF, JC and GA provided administrative and operational support. The initial manuscript was drafted by TD; all authors critically revised the manuscript and approved its final version. DF made a substantial contribution to the revision of the manuscript. UB acts as guarantor. The corresponding author attests that all listed authors meet authorship criteria and that no others meeting the criteria have been omitted.

**Funding** This study was supported by the NIHR Biomedical Research Centre at University Hospitals Bristol and Weston NHS Foundation Trust and the University of Bristol and the British Heart Foundation (Grant No. SP/19/7/34810). All authors had full access to all of the data (including statistical reports and tables) in the study and can take responsibility for the integrity of the data and the accuracy of the data analysis.

**Disclaimer** The funders had no role in the study design, data collection and analysis, decision to publish, or preparation of the manuscript.

**Map disclaimer** The inclusion of any map (including the depiction of any boundaries therein), or of any geographic or locational reference, does not imply the expression of any opinion whatsoever on the part of BMJ concerning the legal status of any country, territory, jurisdiction or area or of its authorities. Any such expression remains solely that of the relevant source and is not endorsed by BMJ. Maps are provided without any warranty of any kind, either express or implied.

**Competing interests** None declared.

**Patient consent for publication** Not applicable.

**Ethics approval** This study does not involve human participants.

**Provenance and peer review** Not commissioned; externally peer reviewed.

**Data availability statement** Data are available in a public, open access repository. All the data used in the study are available from public resources. The dataset is found in the supplemental materials document and can also be made available on request from the corresponding author.

**ORCID iD**
Tim Dong http://orcid.org/0000-0003-1953-0063

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
