## [Reviewer comments · BMJ Open]

ARTICLE DETAILS

TITLE (PROVISIONAL)	A Deep Recurrent Reinforced Learning model to compare the efficacy of targeted local vs. national measures on the spread of COVID-19 in the UK
AUTHORS	Dong, Tim; Benedetto, Umberto; Sinha, Shubhra; Fudulu, Daniel; Dimagli, Arnaldo; Chan, Jeremy; Caputo, Massimo; Angelini, Gianni

VERSION 1 – REVIEW

REVIEWER	Michał Wieczorek Silesian University of Technology, Faculty of Applied Mathematics
REVIEW RETURNED	23-Jan-2021

GENERAL COMMENTS	Authors have prepared a clear description of the results, however there are some missing information which could be beneficial to the overall validation. For example in the previous sections authors have written about a small MAE value, however this topic was not enough continued in the Results section. Authors could also explain the use of RMSprop as the main optimization algorithm. A comparison table between more algorithms could give the reader a better understanding.
---

REVIEWER	Costas Varotsos National and Kapodistrian University of Athens
REVIEW RETURNED	24-Jan-2021

GENERAL COMMENTS	The subject of the paper is very interesting. The results obtained are promising. However authors must elaborate better the methods described, the outcomes obtained and the statistics employed. Additionally, a literature survey in the field is necessary inserting more references, such as: A new model for the spread of COVID-19 and the improvement of safety. Safety Science, 132, 104962, 2020 (https://doi.org/10.1016/j.ssci.2020.104962). Diagnostic Model for the Society Safety under Covid-19 Pandemic Conditions. Safety Science, 105164, 2021 (https://doi.org/10.1016/j.ssci.2021.105164). By summarizing, the results obtained are significant and merit publication after the minor revisions suggested above.
---

REVIEWER	Rajvikram Madurai Elavarasan Texas A&M University
REVIEW RETURNED	11-Apr-2021

GENERAL COMMENTS	The authors have dealt with an interesting topic and it's needed for the Covid scenario. I have few comments to be addressed. Hence I
---

	recommend Major Revision. Abstract must be revised more clearly expressing the Contributions. Introduction must be cited with few of the important literatures related to COVID 19 as mentioned below: https://www.sciencedirect.com/science/article/abs/pii/S0048969720323755 https://www.sciencedirect.com/science/article/abs/pii/S0306261920312290 https://link.springer.com/article/10.1007/s13205-020-02382-3 https://www.sciencedirect.com/science/article/abs/pii/S0959652621002286 https://www.sciencedirect.com/science/article/pii/S2211379721000048 https://www.sciencedirect.com/science/article/abs/pii/S2210670721000810 There can be one section called Research gap which can be included. Conclusions must be improved. Paper must be proof read by an English language expert.
--	--

REVIEWER	Oscar Castillo Tijuana Institute of Technology Department of Computer Science, Graduate Studies
REVIEW RETURNED	07-Aug-2021

GENERAL COMMENTS	The authors of the paper describe their proposed approach for A Deep Recurrent Reinforced Learning model to compare the efficacy of targeted local vs. national measures on the spread of COVID-19 in the UK. The topic is interesting and with possible applicability. However, the paper needs several improvements:  1) the main contribution and originality should be explained in more detail 2) the motivation of the approach with NNs needs further clarification 3) discussion of related work in COVID-19 should be expanded with more recent work 4) Minor grammar and syntax issues need correction 5) more details on the selection of parameters for the neural network should be provided 6) the conclusions should be extended with more possible future work 7) More references to COVID-19 papers should be included (only 9 references, this is low for a scientific paper), like: Multiple ensemble neural network models with fuzzy response aggregation for predicting COVID-19 time series: the case of Mexico. Healthcare 2020;8:181. Modeling COVID-19 epidemic in Heilongjiang province, China, Chaos Solitons Fractals, 138, 1–5. Modeling and forecasting of epidemic spreading: the case of Covid-
--

	19 and beyond. Chaos Solitons Fractals 2020;135:109794. Forecasting of COVID-19 time series for countries in the world based on a hybrid approach combining the fractal dimension and fuzzy logic. Chaos, Solitons and Fractals 140 (2020) 110242 A Novel Method for a COVID-19 Classification of Countries Based on an Intelligent Fuzzy Fractal Approach. Healthcare 2021, 9, 196 A new fuzzy fractal control approach of non-linear dynamic systems: The case of controlling the COVID-19 pandemics, Chaos, Solitons and Fractals, 2021, 151, 111250
--	---

VERSION 1 – AUTHOR RESPONSE

Reviewer: 1

Dr. Michał Wieczorek, Silesian University of Technology

Comments to the Author:

Authors have prepared a clear description of the results, however there are some missing information which could be beneficial to the overall validation. For example in the previous sections authors have written about a small MAE value, however this topic was not enough continued in the Results section.

Response:

Thank you for your positive feedback and helpful suggestions as to how we might improve the manuscript. Additional validation metrics have been added in Model Validation section: Root Mean Squared Error (RMSE), Mean Absolute Error (MAE), Mean Absolute Percentage Error (MAPE), Correlation Coefficient and Average ranking of performance metrics. Results for these metrics are now available in Supplementary Materials, Part II. B Table S2., Table S3. This is now commented on and referred to in the RESULTS section.

Reviewer comment:

Authors could also explain the use of RMSprop as the main optimization algorithm. A comparison table between more algorithms could give the reader a better understanding.

Response:

Selection rationale for RMSProp and a comparison table with other optimization algorithms is now available in the Supplementary Material, Part I Recurrent Optimization Algorithms. This is also referred to in the main text.

Reviewer: 2

Dr. Costas Varotsos, National and Kapodistrian University of Athens Comments to the Author:

The subject of the paper is very interesting. The results obtained are promising.

However authors must elaborate better the methods described, **the outcomes obtained** and the statistics employed.

Response:

Thank you for providing this positive feedback on our manuscript, and for your encouraging comments on the results obtained. The methodological sections have been revised extensively to increase clarity and to signpost to the supplementary materials section whereby possible. The word count recommendation for BMJ Open is 4000 words which is challenging when reporting research findings. The word count for the paper when submitted was 3926 words, but this has increased to accommodate extra material that has been requested the reviewers. To keep the manuscript with the journal's word limit boundaries, it is essential that we place some of the methodological details in the supplementary materials.

Additional validation metrics have been added in Model Validation section: Root Mean Squared Error (RMSE), Mean Absolute Error (MAE), Mean Absolute Percentage Error (MAPE), Correlation Coefficient and Average ranking of performance metrics. Results for these metrics are now available in Supplementary Materials, Part II. B Table S2., Table S3. This is now commented on and referred to in the RESULTS section.

Reviewer comment:

Additionally, a literature survey in the field is necessary inserting more references, such as:
A new model for the spread of COVID-19 and the improvement of safety. Safety Science, 132, 104962, 2020 (<https://doi.org/10.1016/j.ssci.2020.104962>).
Diagnostic Model for the Society Safety under Covid-19 Pandemic Conditions. Safety Science, 105164, 2021 (<https://doi.org/10.1016/j.ssci.2021.105164>).
By summarizing, the results obtained are significant and merit publication after the minor revisions suggested above.

Response:

Thank you for these useful references, they are very interesting and have now been cited in the introduction section. Thank you once again for your positive recommendation.

Reviewer: 3

Dr. Rajvikram Madurai Elavarasan, Texas A&M University

Comments to the Author:

The authors have dealt with an interesting topic and it's needed for the Covid scenario. I have few comments to be addressed. Hence I recommend Major Revision.

Response:

Thank you for your positive comments on this topic. We shall address your comments below.

Reviewer comment:

Abstract must be revised more clearly expressing the Contributions.

Response:

The abstract has now been extensively revised to highlight contributions and results updated to better reflect statistical performance. The purpose of the study is more clearly defined in the objective. Further work is suggested in conclusion.

Reviewer comment:

Introduction must be cited with few of the important literatures related to COVID 19 as mentioned below:

<https://www.sciencedirect.com/science/article/abs/pii/S0048969720323755>

<https://www.sciencedirect.com/science/article/abs/pii/S0306261920312290>

<https://link.springer.com/article/10.1007/s13205-020-02382-3>

<https://www.sciencedirect.com/science/article/abs/pii/S0959652621002286>

<https://www.sciencedirect.com/science/article/pii/S2211379721000048>

<https://www.sciencedirect.com/science/article/abs/pii/S2210670721000810>

Response:

Thank you for these references and we believe these are tremendously valuable work that should be incorporated. These have now been cited in the introduction section.

Reviewer comment:

There can be one section called Research gap which can be included.

Response:

A research gap section has now been added in the section below the introduction.

Reviewer comment:

Conclusions must be improved.

Response:

Thank you for this helpful suggestion. We have extensively revised the conclusion and discussion. Both these sections are now extended with discussion on possible future work. The conclusion is now more focused and summarise the main contributions of this work.

Reviewer comment:

Paper must be proof read by an English language expert.

Response:

A clinician and lecturer with expertise in literature review and writing was added to the co-authoring team to revise and proof-read the manuscript. In addition, the entire manuscript is revised again and proof-read by all existing co-authoring team members.

Reviewer: 4

Prof. Oscar Castillo, Tijuana Institute of Technology Department of Computer Science

Comments to the Author:

The authors of the paper describe their proposed approach for A Deep Recurrent Reinforced Learning model to compare the efficacy of targeted local vs. national measures on the spread of COVID-19 in the UK. The topic is interesting and with possible applicability.

Response:

Thank you for your positive comments on this topic and its potential applicability. We shall address your comments below.

Reviewer comment:

However, the paper needs several improvements:

1) the main contribution and originality should be explained in more detail

Response:

Thank you for this important comment. We have added a RESEARCH GAP section which describes the current gaps in this topic and how this work aims to address these limitations.

Reviewer comment:

2) the motivation of the approach with NNs needs further clarification

Response:

The motivation for this approach of NNs is now addressed in the introduction section and also RESEARCH GAP section.

Reviewer comment:

3) discussion of related work in COVID-19 should be expanded with more recent work

Response:

Thank you for this helpful comment. We have increased the number of references considerably (n=32). Many of these are now referred to in the discussion.

Reviewer comment:

4) Minor grammar and syntax issues need correction

Response:

The manuscript has been revised to increase clarity and sentence structures.

Reviewer comment:

5) more details on the selection of parameters for the neural network should be provided

Response:

In the Model Development section, we have now added signpost to refer to the Supplementary Material, Part I Neural Network architecture and configuration for detailed specification of model parameters. A Recurrent Optimization Algorithms section has been added in the Supplementary Materials to provide further rationale for selecting RMSProp as the optimization algorithm. The word count recommendation for BMJ Open is 4000 words which is challenging when reporting research findings. The word count for the paper when submitted was 3926 words, but this has increased to accommodate extra material that has been requested the reviewers. To keep the manuscript within the journal's word limit boundaries, it is essential that we place some of the methodological details in the supplementary materials.

Reviewer comment:

6) the conclusions should be extended with more possible future work

Response:

Thank you for this helpful suggestion. We have extensively revised the conclusion and discussion. Both these sections are now extended with discussion on possible future work.

Reviewer comment:

7) More references to COVID-19 papers should be included (only 9 references, this is low for a scientific paper), like:

Multiple ensemble neural network models with fuzzy response aggregation for predicting COVID-19 time series: the case of Mexico. Healthcare 2020;8:181.

Modeling COVID-19 epidemic in Heilongjiang province, China, Chaos Solitons Fractals, 138, 1–5.

Modeling and forecasting of epidemic spreading: the case of Covid-19 and beyond. Chaos Solitons Fractals 2020;135:109794.

Forecasting of COVID-19 time series for countries in the world based on a hybrid approach combining the fractal dimension and fuzzy logic. Chaos, Solitons and Fractals 140 (2020) 110242

A Novel Method for a COVID-19 Classification of Countries Based on an Intelligent Fuzzy Fractal Approach.

Healthcare 2021, 9, 196 A new fuzzy fractal control approach of non-linear dynamic systems: The case of controlling the COVID-19 pandemics, Chaos, Solitons and Fractals, 2021, 151, 111250

Response:

Thank you for these useful references, they are very interesting and have now been cited in the introduction section. The word count recommendation for BMJ Open is 4000 words which is challenging when reporting research findings. The word count for the paper when submitted was 3926 words, but this has increased to accommodate extra material that has been requested the reviewers.

VERSION 2 – REVIEW

REVIEWER	Oscar Castillo Tijuana Institute of Technology Department of Computer Science, Graduate Studies
REVIEW RETURNED	03-Dec-2021
GENERAL COMMENTS	The authors have addressed all my concerns and the paper can be accepted.